# Serum Paraprotein Is Associated with Adverse Prognostic Factors and Outcome, across Different Subtypes of Mature B-Cell Malignancies—A Systematic Review

**DOI:** 10.3390/cancers15184440

**Published:** 2023-09-06

**Authors:** Maria Christina Cox, Fabiana Esposito, Massimiliano Postorino, Adriano Venditti, Arianna Di Napoli

**Affiliations:** 1UOC Malattie Linfoproliferative, Fondazione Policlinico Tor Vergata, 00133 Roma, Italy; 2Ematologia, Dipartimento di Biomedicina e Prevenzione, Università Tor Vergata, 00133 Roma, Italy; fabiana.e91@gmail.com (F.E.);; 3Department of Clinical and Molecular Medicine, School of Medicine and Psychology, Sapienza University, 00189 Roma, Italy; arianna.dinapoli@uniroma1.it

**Keywords:** non-Hodgkin lymphoma, NHL, diffuse large B-cell lymphoma, follicular lymphoma, marginal-zone lymphoma, MALT, EMZL, chronic lymphocytic leukemia, paraprotein*, gammopathy, monoclonal immunoglobulin, M-protein, free light chain, prognostic marker, outcome, survival, central nervous system

## Abstract

**Simple Summary:**

This systematic review encompasses the clinical impact of serum paraprotein (PP), in low- and high-grade mature B-cell malignancies (except for myeloma and Waldenstrom Macroglobulinemia). In fact, reports on this topic are sparse and heterogeneous, and an overall view is lacking. Literature analysis shows that, across different entities, PP is consistently associated with high-risk biological and clinical features, as well as with poor outcome. Indeed, screening for serum PP should be included in the diagnostic work-up of all mature B-cell malignancies.

**Abstract:**

The presence of a serum paraprotein (PP) is usually associated with plasma-cell dyscrasias, Waldenstrom Macroglobulinemia/lymphoplasmacytic lymphoma, and cryoglobulinemia. However, PP is also often reported in other high- and low-grade B-cell malignancies. As these reports are sparse and heterogeneous, an overall view on this topic is lacking, Therefore, we carried out a complete literature review to detail the characteristics, and highlight differences and similarities among lymphoma entities associated with PP. In these settings, IgM and IgG are the prevalent PP subtypes, and their serum concentration is often low or even undetectable without immunofixation. The relevance of paraproteinemia and its prevalence, as well as the impact of IgG vs. IgM PP, seems to differ within B-NHL subtypes and CLL. Nonetheless, paraproteinemia is almost always associated with advanced disease, as well as with immunophenotypic, genetic, and clinical features, impacting prognosis. In fact, PP is reported as an independent prognostic marker of poor outcome. All the above call for implementing clinical practice, with the assessment of paraproteinemia, in patients’ work-up. Indeed, more studies are needed to shed light on the biological mechanism causing more aggressive disease. Furthermore, the significance of paraproteinemia, in the era of targeted therapies, should be assessed in prospective trials.

## 1. Introduction

The diseases classically associated with paraproteinemia include monoclonal gammopathy of undetermined significance, multiple myeloma, Waldenstrom Macroglobulinemia/lymphomplasmacytic lymphoma (MW/LPL), cryoglobulinemia, and light chain amyloidosis. Nonetheless, in recent years, several papers have reported that almost all B-cell non-Hodgkin lymphoma (B-NHL) subtypes, chronic lymphocytic leukemia (CLL), and monoclonal B-cell lymphocytosis (MBL) may secrete PP. Furthermore, NHL and CLL associated with gammopathy, appear to have distinct clinicopathological features, and outcome, compared to those without a serum PP. However, these reports are sparse and heterogeneous, and an overall view on this topic is lacking; therefore, this paper will focus on the features of CLL and NHL (other than WM/LPL) associated with gammopathy. The capability to secrete immunoglobulin (Ig) is a finely regulated process in the development of B-cells, which is acquired by terminally differentiated lymphocytes. In fact, the progression from a mature B-cell to an antibody-secreting cell (ASC), occurs following the adaptive immune response to an antigen, by silencing the B-cell program and activating the ASC one [1,2,3]. During normal B-cell maturation, chromosome recombination of the V (variable), D (diversity), and J (junctional) gene segments forms the V region of the heavy (IGH) and light (Igk/Igλ) immunoglobulin chains. The region of the rearranged gene, where the three genomic segments (VH-DH-JH) are joined, is the third complementarity determining region (CDR3) of the heavy chain, which is the most variable portion of immunoglobulin (Ig) molecules. When a mature B-cell encounters an antigen, further diversification occurs through somatic hypermutation (SHM) of the rearranged IGHV gene, and class-switching recombination (CSR). Also, SHM introduces random nucleotide changes in the V genes, so that B-cells, that produce immunoglobulins with the highest degree of affinity, are selected (B-cell receptor affinity maturation) [4], whereas CSR leads to gene substitution of the immunoglobulin heavy constant region, switching antibody production from IgM to IgG, IgE, or IgA [5]. These processes occur in the germinal centers of secondary lymphoid organs. Since the germline genes used to encode the V region have been mapped, it is possible to determine whether the malignant B-cell clone has undergone somatic hypermutation [6].

When an immunoglobulin is secreted by an abnormal B-cell clone, this represents a serum paraprotein (PP) [7]. Indeed, PP may be a whole, free light, or heavy chain, or a fragment thereof [8]. The routine screening tests used in clinical practice to detect paraproteinemia (PP) in serum and urine are: protein electrophoresis (PE) and immune-electrophoresis (IFE), or the hevylite chain assay (HLC, hevylite^TM^, the Binding site, Birmingham UK), plus the serum free-light chain test (sFLC, Freelite^TM^, the Binding site, Birmingham UK). The latter allows to quantify k and λ light chains’ concentration and the k/λ ratio, which are useful for both disease prognostication and monitoring. In addition, sFLC is pivotal to diagnose light chain diseases, However, the latter cannot stand alone in the diagnostic work-up of PP. In fact, in ≥30% of patients with paraproteinemia, there is a normal k/λ ratio [9], while an abnormal k/λ chain ratio is present in up to 36% of subjects, without monoclonal lymphoproliferation [9,10].

For all of the above, we did not include in our analysis studies which relied only on the sFLC test [11,12,13,14,15,16].The definitve aim of this manuscript is to prompt a better understanding of B-cell neoplasms associated with gammopathy, and possibly to contribute to the refinement of NHL classification.

## 2. Methods and Criteria of Bibliographic Research and Articles Selection

We performed a literature search focusing on B lymphoproliferative disorders and their association with serum PP. **Inclusion criteria:** adult patients; B-cell lymphoproliferative diseases, retrospective and prospective series, clinical trials, and publication years 2005–2023; **Exclusion criteria:** reviews, case reports, papers devoid of data on treatment and outcome of patients, meeting abstracts, papers not edited in the English language, reports on WM/LPL, T-cell lymphoma, Hodgkin lymphoma, and studies which assessed the presence of PP only by sFLC analysis. Search methods and engines: PUBMED, Web of Science, and Cochrane databases were interrogated for the terms: “paraprotein*”, “gammopathy”, “monoclonal component”, “free light chain”, “freelite” AND “chronic lymphocytic leukaemia “OR“ “non-Hodgkin lymphoma”. In addition, we also analyzed articles dealing with prognostic factors and scores in NHL and CLL, to search for additional data. This study has not been registered in the Prospero website.

## 3. Search Results

A total of 1589 studies were examined for titles and abstracts; 1559 studies were excluded, based on the inclusion and exclusion criteria (Figure 1). Finally, 30 papers were examined as full-text, and 9 of these were excluded due to: (1) lack of essential information, or (2) the assessment of monoclonal paraprotein was carried out only by the FLC assay. Furthermore, 5 more articles were found in the bibliography of retrieved articles. Overall, 26 articles were selected and thoroughly analyzed for this systematic review.

## 4. Paraprotein-Associated Mature B-Cell Lymphoproliferative Diseases

### 4.1. Chronic Lymphocytic Leukaemia

Chronic lymphocytic leukemia (CLL) is characterized by a very heterogeneous clinical course with survival ranging from a few years to decades [17]. Indeed, the marked heterogeneity of patients’ outcome is related to specific biological and clinical features of the disease [18]. Among the factors associated with overall survival (OS) are: age > 65 years, Rai stage I-IV, serum beta-2 microglobulin > 3.5 mg/L, unmutated IGVH genes, del17p, or TP53 gene mutation, which identify four risk groups with five-year OS ranging from 93% (low-risk patients) to 23% (high-risk patients) [19].

One hypothesis to explain the worse prognosis of unmutated IGHV (uIGVH) CLL, could be that neoplastic cells are less prone to undergo apoptosis when exposed to conventional R-Chemotherapy (RCHEMO). Furthermore, whole-exome sequencing of CLL samples identified a higher frequency of driver mutations in uIGHV than in IGHV-mutated CLL [5]. So far, other prognostic factors, such as trisomy 12 (+12), del11q, del13q, BIRC3, SF3B1, NOTCH1, have been identified [18].

CLL was reported to be the lymphoproliferative disease with the highest prevalence of serum paraproteins (after LPL/WM) [20]. However, only in the last decade, paraproteinemia was associated with adverse biological and clinical features. Xu et al. analyzed the impact of paraproteinemia in a series of 27/133 (20%) consecutive CLL patients diagnosed between 2004 and 2010 [21]. Serum PP was strongly correlated with advanced stage and other poor prognostic factors, including del17p/TP53mut. In univariate analysis, PP was predictive for shorter OS, while in multivariate analysis, only IgM PP was associated with poor survival (Table 1).

In 2015, Rizzo et al. [22] investigated a series of 150 consecutive newly diagnosed CLL. Gammopathy was present in 52 (34.6%) cases: 27 IgM and 25 IgG, while 41 (29%) had hypo-γ. Binet stage C, del 17p, and usage of IGHV3-48, IGHV3-74, and IGHV6-1 genes were overrepresented in the IgM paraprotein (IgM PP+ve) subgroup. The IgG PP+ve subset had increased frequencies of IGHV3-48 and IGHV4-39 rearrangements, and +12. The hypo-γ group more frequently exhibited the IGHV3-21 and IGHV3-23 genes usage, while the normal-gammaglobulin subset was characterized by an over-representation of IGHV1-2 genes rearrangement, and a marked tendency to be associated with long CDR3 segments. The SF3B1 gene mutation clustered in the PP+ve and in the hypo-γ subgroups [22]

Of note, the PP+ve subjects, showed no prevalence of mutated or uIGVH. Interestingly, solely in the IgM PP+ve subset, there were no differences in outcome between IGVH mutated and unmutated patients. In multivariate analysis (MV), only IgM or IgG PP, hypo-γ and uIGVH were predictive for shorter treatment-free survival (TFS). Subsequently, Corbingi et al. [23] in a retrospective study, divided a cohort of 1505 CLL patients, diagnosed between 1987 and 2016, into four groups: (1) 149 (9.9%) IgG+ve, (2) 73 (4.8%) IgM+ve, (3) 200 (13.3%) hypo-γ, and (4) 1083 (72%) with normal Ig levels and no PP. IgMs-CLL was correlated with advanced stage and del17p or TP53 mutations, while trisomy 12 was more frequently detected in the IgG+ve CLL subset. In the latter, the IGHV gene repertoire was made up of IGHV 1–69, 4–59 and 3–33. Likewise, IgM+ve CLL showed a distinct IGVH repertoire, with the prevalent usage of IGHV 1-03, 3-09 and 3-43. The median patients’ follow-up was 78 months (range 1–333). In MV analysis, the independent factors predictive for shorter TFS were: uIGHV (*p* < 0.0001), Binet stage B or C (*p* < 0.0001 and *p* = 0.0002, respectively), del11q (*p* < 0.0001), hypo-gamma (*p* < 0.0001), and IgG or IgM PP+ve (*p* = 0.001 and *p* < 0.0001, respectively). (Table 1). Meanwhile, independent factors for OS were only age ≥65 (*p* = 0.005), del17p/TP53 mutation (*p* = 0.02), Binet stage B/C (*p* < 0.0001 and *p* = 0.005, respectively), and IGVH mutational status (*p* < 0.0001) [23]

### 4.2. Diffuse Large B-Cell Lymphoma

Diffuse large B-cell lymphoma (DLBCL) accounts for about one-third of NHL in the western world and it is the most common histological subtype [24]. Indeed, DLBCL is a phenotypically, genetically, and clinically heterogeneous entity. Cell of origin (COO) analysis based on gene-expression-profiling analysis, allowed the identification of: activated B-cell type (ABC-DLBCL), germinal center B-cell-type DLBCL (GCB-DLBCL), and unclassified (U-DLBCL) subtypes. GCB and ABC subtypes, reflect the state of B-cell differentiation and maturation. ABC-DLBCL is characterized by SHM of the IGHV gene, and by a significantly higher number of Ig-derived neoantigens. DLBCL cells mostly express surface immunoglobulins: the ABC subgroup has a higher incidence of IgM expression, whereas the GCB subgroup, generally expresses IgG and IgA [25].

ABC- and GCB-DLBCL subgroups have diverse genetic abnormalities which target diverse survival pathways [26]. ABC-DLBCL is featured by chronically active B-cell receptor (BCR) signaling, which leads to a constitutive activation of NF-κB. Conversely, in the GCB-DLBCL subset, there is an activation of the phosphoinositide-3-kinase–protein kinase (PI3K) pathway signaling [27,28] Very recent studies using modern molecular technologies have attempted to further separate the genetic subtypes of DLBCL [29,30,31]. However, despite new knowledge and improved biological understanding, they still have limited impact on everyday clinical practice [32]. Hence, interest has arisen for new prognostic factors, such as serum markers; Jardin et al. [33] in 2013, investigated the presence of paraprotein in 409 DLBCL patients enrolled in different GELA trials, using both the sFLC and the sHLC technologies (Table 1). The median follow-up was 42.9 months and 92% were treated with R-CHOP-like schedules. Serum PP subtypes were: IgA in 2.9%, IgM in 19.1%, and IgG in 6.4% samples, respectively [33]. An abnormal k/λ ratio was observed overall in 9.3% of samples and in only one-third of subjects with a monoclonal IgM. Seventeen patients (4.2%) displayed an elevation of both k and λ FLCs, and 76 (18.6%) patients displayed an elevation of k or k/λ FLCs. Among 16 patients with an abnormal IgMk/IgM λ ratio, 5 were classified as GCB-type and 11(69%) as ABC-type or unclassifiable (*p* = 0.14, Fisher’s exact test). The 78 patients, with an abnormal IgMk/IgMλ ratio, had inferior PFS and OS compared to patients without it, both in the overall cohort (5y-PFS CI95% 44.9% vs. 69.3%, *p* = 0.0003 and 5y-0S CI95% 50.8% vs. 78.1%, *p* = 0.0003) and in the R-CHOP cohort (5y-0S 43.5% vs. 70.3%, *p* = 0.003). In MV analysis, including IPI or the five IPI factors, an abnormal IgMk/IgMλ ratio (HR = 1.54, CI95% 1.03–2.3, *p* = 0.032) remained predictive of worse PFS. Conversely, abnormal IgGk/IgGλ and IgAk/IgAλ ratios were not predictive of poor outcome. Maiolo et al. also analyzed HLC and FLC in 106 DLBCL patients with (*n* = 56) and without (*n* = 45) serum PP. They showed that: (1) only IgM, and not other PPs, was predictive of poor outcome; (2) both PP+ve and PP-ve could have higher than normal levels of FLC; Noteworthy, higher levels of FLC were predictive of shorter PFS, only in the PP-ve subgroup [34].

Cox et al. reported a retrospective series of 151 DLCL, and 17 (12.5%) of them were defined as IgM-secreting (IgMs-DLBCL), since there was concordance between the heavy and light chains of the serum PP with those expressed by the tumor cells in 89% [35]. IgMs-DLBCL was characterized by older age, advanced disease, high LDH, non-GCB type, bone-marrow infiltration, and extensive extra-nodal localization. In addition, IgMs-DLBCL patients had a higher frequency of secondary central nervous system (CNS) localization. In MV analysis, IgMs was an independent prognostic factor for both PFS and OS. Of note, IgM PP faded rapidly after treatment starting, and, at relapse, IgM PP was detectable in only in a minority of cases.

In 2022, the same group published a large multicenter study of 158 newly diagnosed DLBCL associated with gammopathy. Of these, 102 (64.5%) patients harbored a monoclonal IgM, and 56 (34.5%) had other PPs (IgG = 53, IgA = 3, other = 1). The median serum PP value was 9.2g/L (range not quantifiable–29.4 g/L) [36].

Histological and clinical features, and outcome were compared with a consecutive series of 492 PP-ve. Furthermore, molecular and immunogenetic features were investigated in a subset of 25 IgMs-DLBCL. This larger study confirmed the previously reported adverse clinical features, as well as IgM PP is an independent prognostic factor for both PFS and OS (*p* < 0.001). Furthermore, PPs other than IgM, were not associated with adverse prognostic factors and outcome. Of note, in this large cohort of IgMs-DLBCL, it was confirmed a high incidence of CNS spreading compared to PP-ve cases (15% vs. 2% *p* < 0.0001), [35]. Overall, histological, and molecular analyses revealed that IgMs-DLBCL had: (1) recurrent mutations of TP53 and/or MYD88 and/or CD79B genes in 60% of analyzed subjects; (2) prevalence of non-GCB/ABC-type; (3) preferential IGVH4-34 gene usage; and (4) frequent BCL2 overexpression. As IgMs-DLBCL is a sizable subset with a dismal prognosis, the authors recommend that immunofixation should be included in the pre-treatment assessments. More recently, Johansson et al. reported IgM PP and interim PET to be the most powerful indicators in MV of both PFS and OS in a series of patients enrolled in the PETAL study [37]. Unfortunately, only 108/609 (18%) patients could be investigated for IgM gammopathy by both sFLC and sHLC. Nonetheless, in the 19 (18%) cases identified, adverse prognostic features and a high incidence of CNS involvement at relapse (16.6% vs. 0%) were reported. Fifteen 15 IgM+ve cases were analyzed for gene mutations and COO. However, results failed to show the prevalence of the ABC-type and of MYD88 and TP53 genes’ mutations [38].

Other authors, almost from the far East, reported that DLBCL associated with gammopathy has adverse prognostic features and has mostly a non-GCB-type phenotype. Furthermore, all authors reported gammopathy to be a strong and independent prognostic factor for both PFS and OS. However, at odds with authors from western countries, no relevant differences were described between IgM and other paraproteins [39,40,41,42].

### 4.3. Follicular Lymphoma

In 2021, Mozas P et al. reported a series of 311 consecutive FL patients, with grade 1–3A, who were investigated for the presence of serum PP [43]. Primary cutaneous, primary gastro-intestinal, and grade 3B FL were not included in the study. PP was detected in 82 (26%) subjects, who, compared to PP-ve, more frequently showed an elevated B2-microglobulin (*p* = 0.008), and also a slightly older median age (60 vs. 61.5, *p* = 0.002), but not a prevalence of high-risk FLIPI score. Of note, the rate of PP+ve subjects increased with age, as it was 12% in patients <39 years-old and 48% in those >80 years-old. The median concentration of PP was 2.7 g/L (range, from not detectable–39 g/L). However, the majority of PPs were detected only by immunofixation, as their serum concentration was very low. Only in 39 cases the type of PP was known: 49% IgG, 26% IgM, 5% IgA, 11% free-light chains, and 9% biclonal PP, respectively. When FL cells were analyzed for the expression of heavy and light chains, concordance with the serum PP was detected in 68%. During the follow-up period, which for surviving patients was of 4.6 years (median), 279 (89%) subjects received systemic treatment (87% R-CHEMO, 13% R monotherapy). Progression of disease within 24 months (POD24) was significantly associated with the presence of PP (27% vs. 15%, *p* = 0.02). Although ten-year OS and PFS were significantly shorter for the PP+ve subset, in the MV model this factor lost its significance (Table 1). Notwithstanding, it remained significant in patients ≥60 years (HR = 2.4, *p* = 0.02) (Table 1) In fact, gammopathy, B symptoms, and Ecog PS > 1 were the only negative prognostic factors in this model.

### 4.4. Marginal Zone Lymphoma

Marginal zone lymphomas (MZL) are the third most common type of B-cell non-Hodgkin lymphoma. This heterogeneous group of malignancies comprises four main subtypes: extranodal marginal zone lymphoma of mucosa-associated lymphoid tissue (EMZL), primary cutaneous marginal zone lymphoma (PCMZL), nodal marginal zone lymphoma (NMZL), and Pediatric NMZL (PNMZL). Splenic marginal zone lymphoma (SMZL) is now considered under the umbrella of splenic lymphomas [24]. Furthermore, non-CLL-type monoclonal B lymphocytosis with features consistent with a marginal zone origin (CBL-MZ) is considered as a premalignant condition [44]. MZLs are thought to originate from the transformation of a mature memory B-cell normally present in the marginal zone of secondary lymphoid follicles. It usually has an indolent course, and the heterogeneous clinical features depend on the organ involved.

#### 4.4.1. Extranodal Marginal Zone Lymphoma

Most data on gammopathy in MZL are derived from EMZL. In fact, a high rate of PP+ve cases has been reported in this subgroup by several studies. EMZL is characterized by genetic abnormalities involving the activation of NF-κB, through deregulation of a signaling complex comprising several genes (CARD11, BCL10, and MALT1). Among the translocations involved, there are those affecting the IGH locus on chromosome 14—t(1;14)(p22;q32) and t(14;18)(q32;q21)—leading to up-regulation of BCL10 and MALT1, respectively. The FOXP1/IGH translocation is additionally present in 10% of EMZL [45]. Large published series reported paraproteinemia in from 10.7% up to 39% of patients. Wöhrer et al. [46] in 2004, described that in 19 out of 52 patients (36%) (31 were extragastric EMZL), the detection of a PP (IgM = 12; IgG = 6, IgA = 1; K-chain = 8, λ-chain = 5). Gammopathy was not associated with HP infection, t(11,18), or with advanced clinical stage. However, at odds with PP-ve patients, with HP+ve gastric EMZL, subjects who harbored a PP, did not respond to HP eradication, [46]. In the same year, another study by the George Town University reported a serum PP in 7/26 (27%) patients (IgG = 5, IgM = 1, IgA = 1) [47]. In the latter report, PP was associated with more advanced disease (*p* = 0.007) and bone marrow involvement. Both authors reported that PP concentration may be undetectable on protein electrophoresis and they suggest that both immune-electrophoresis and immunofixation should be used in the work-up of patients to rule-out PP+ve cases. In 2006, Arcaini et al. [48] published a series on non-gastric EMZL, in which 36/208 (17%) were PP+ve (IgM = 67%, IgG = 25% and IgA = 8%). This subset had advanced disease and older age. Ninety-four patients received chemotherapy or polychemotherapy (49%) up-front, and only five of them received rituximab. Of note, in MV analysis of OS, PP+ve was a significant negative prognostic factor (*p* = 0.04), as well as bone marrow involvement (*p* = 0.004), distant/diffuse nodal involvement (*p* = 0.01), ECOG score ≥ 2 (*p* = 0.007), and Hb < 11 g/dl (*p* = 0.04) [48]

More recently, Alderuccio et al.[49] reported a large cohort of EMZL patients. In total, 35 out of 328 (10.7%) cases harbored a PP (IgG in 43%, IgM in 37%, IgA in 14%, and lambda light chain only in 3%). This was more frequently detected in patients with (1) gastro-intestinal, other than gastric, involvement; (2) non-GI EMZL; and (3) multiple-site disease (17.9%). Initial treatment was radiation alone (n = 210, 51.9%), chemotherapy (n = 114, 28.1%), combined chemotherapy and radiation (n = 36, 8.9%), and surgery (n = 22, 5.4%). Overall, 150 patients received systemic regimens, which, in 90% of cases, was R-CHEMO. Univariate analysis showed that PP was associated with both shorter PFS and OS. Conversely, in MV analysis, the only factors which had a significant association with shorter PFS were: >60 years, elevated LDH, multiple site involvement, and non-CR after initial therapy. While factors for OS were: age > 60 year, anemia, multiple extranodal site involvement, non-CR after initial therapy, and high-grade transformation. The authors speculated that PP lost significance in MV, either because PP+ve cases were overall few, or because they were associated with other adverse features [49].

In 2022, also Ren and colleagues evaluated in a consecutive series of 218 patients, diagnosed between 2001 and 2021, whether serum PP could be a prognostic factor for PFS and OS in patients with EMZL lymphoma [50].

PP was detected in 42/218 (19.3%) subjects, and it was significantly associated with adverse prognostic factors, such as: age ≥ 70 years, advanced Ann Arbor stage, elevated β2-microglobulin, bone marrow involvement, and high-risk MALT-IPI [51], as well as the need for systemic treatment. The relative incidence of immunoglobulin subtype was: IgM = 64%, IgG = 12%; IgA = 4.9%; biclonal gammopathies = 12%; and Lambda light chain = 2%. Most patients had localized disease and were treated with anti-HP therapy (gastric) and radiotherapy. In total, 103 subjects (47.2%) required systemic treatment (upfront or following progression): 23 received Rituximab monotherapy and 80 received R-chemo. Of note, 74% of patients with PP received systemic treatment versus 41% who did not have paraproteinemia (*p* < 0.001). In univariate analysis, both IgG and IgM PP+ve were associated with shorter OS (*p* = 0.0002 and *p* = 0.0068). Meanwhile, only IgM PP+ve was associated with shorter PFS (*p* = 0.0002). In MV analysis, significant prognostic factors for PFS were: MALT-IPI intermediate (*p* = 0.036) and high (*p* = 0.007), systemic therapy (*p* = 0.038), and PP+ve (*p* = 0.018). Meanwhile, for OS the only significant variables were high MALT-IPI score (*p* = 0.017) and PP+ve (*p* = 0.026). Of not.

#### 4.4.2. Primary Cutaneous Marginal Zone Lymphoma (PCMZL)

There are few data on this subset, which may be associated with gammopathy in up to 39% of cases. Nonetheless, the few existing data support the notion that PP distinguishes a more aggressive disease [52].

#### 4.4.3. Splenic Marginal Zone Lymphoma and CBL-MZ

Serum PP is found in up 42% of patients with SMZL [53,54] or with CBL-MZ [44]. In such cases, screening for the MYD88L265P mutation is mandatory to exclude WM/LPL.

Xochelli et al. [44] retrospectively investigated clinical, morphological, immune-phenotypical immunogenetics, and the molecular and cytogenetic characteristics of 102 patients with CBL-MZ. A total of 27 out of 81 (33%) cases had paraproteinaemia (IgG = 12/25, IgM = 13/25, k-chain = 17/25, λ-chain = 8/25). Following a median follow-up of five years, 17 (16.6%) patients progressed to overt lymphoma. This subgroup frequently showed a complex karyotype, while other features, such as the status of the IGHV or paraproteinemia, were not predictive of progression [44].

In 2002, a series of 81 patients with SMZL were investigated for clinical features and outcome [53]. A serum PP was observed in 34 (42%) patients. This was IgM in 21 (62%), IgG in 7 (21%), IgA in 3 (9%), and IgA + IgM in 2 cases (6%). The k and λ chains were present in 26 and 6 cases, respectively. The median concentration of PP was 6.5 gr/L (range 1–51). PP+ve patients were more frequently affected by autoimmune hemolytic anemia and thrombocytopenia. Overall splenectomy was carried out in 79%, and in 31% this was the only treatment. Conversely, chemotherapy alone (CHOP, Fludarabine, Chlorambucil, ACVB) was administered to 44%, and in 16% this was the sole treatment. Median overall survival (OS) and time to progression (TTP) for the whole group of patients were 10.4 years and 3.7 years, respectively. High levels of Beta2-microglobulin, lymphocytes > 20.000 mcL, and PP+ve were the only factors correlated with shorter OS at five years (paraproteinemia 72% vs. 100%, *p* = 0.006). In MV analysis, the only significant negative prognostic factors for OS were the presence of autoimmune events (*p* < 0.006) and PP (*p* < 0.001). Conversely, in the large series published by Montalban et al. [54]. In 127/501 (25%), PP was correlated with advanced disease and the decision to start treatment (*p* = 0.036), but not with patients’ outcome[54].

Very recently, other two papers describing the same cohort, have further endorsed the relevance of paraproteinemia in MZL, in a series of 524 adult patients with various MZL subtypes (NMZL, EMZL, SMZL), who received first-line treatment from 2010 onward [55,56]. The median follow-up was of 17.2 years. One-hundred-and-one (32%) subjects had paraproteinemia (EMZL = 50%, NMZL = 28%, SMZL = 22%), the predominant type was IgM (56%), followed by IgG (31%) and IgA (3%); biclonal PP and light chain PP was found in 5% each. The level of PP was undetectable in 10% and, in the majority of patients, was ≤15 g/L. The PP+ve group, compared to the PP-ve one, was characterized by older age, worse performance status, advanced stage, and lower albumin. As first-line treatment, R-monotherapy was administered to 45%, R-CHEMO to 28%, other CHEMO to 4%, radiotherapy to 19%, and surgery to 7%. PP+ve patients, compared to PP-ve, were more frequently treated with R-CHEMO compared to other modalities (37% vs. 23%). In addition, in the PP+ve group, transformation to high-grade was significantly more frequent both at three and five years (*p* = 0.001) from diagnosis. Furthermore, in MV logistic regression analysis, gammopathy was the variable mostly correlated with POD24 (47% of cases, HR = 2.87, *p* < 0.0001) [57]. In PP+ve subjects, the 3- and 5-year estimated PFS were 51% (95%CI, 42–59%) and 42% (95%CI, 33–51%), respectively, while in the PP-ve group were 73% (95%CI, 68–78%) and 60% (95%CI,53–66%), respectively, (log-rank *p* < 0.001). After adjusting for factors associated with inferior PFS in univariate models, the PP+ve subset remained significantly associated with inferior PFS in MV analysis (HR = 1.74, 95%CI = 1.20–2.54, *p* = 0.004). However, gammopathy lost its significance in patients treated with R-Bendamustine. The negative impact of PP on PFS was significantly worse in both EMZL and NMZL than in SMZL, but the difference did not reach statistical significance. Conversely, PP+ve did not adversely affect OS. Of note, there was no significant difference in PFS based on the PP subtype and its serum concentration (Table 1).

### 4.5. Other B-Cell Lymphoproliferative Disorders

There are few studies, mostly case reports, on the secretion of PP in the serum of patients affected by other types of B-cell lymphoproliferative malignancies, such as mantle cell lymphoma and hairy cell leukemia [20]. Conversely, Primary-Cold-Agglutinin-Associated B-lymphoproliferative disease is worth a mention. This is a very rare entity characterized by the exclusive usage of the IGVH4-34 gene, which encodes an IgM cold agglutinin that targets I antigen on red blood cells [58]. Recently, it was shown that this malignancy frequently harbors the inactivating mutation of the KMT2D gene, which impedes immunoglobulin class switching, as well as CARD11 gene mutation, involved in the NF-kb pathway [58].

## 5. Discussion

Systematic review of existing published series shows that PP+ve patients, compared to PP-ve ones, have worse clinical and molecular features, which strongly impact clinical outcome Noteworthy, these data are consistent across different mature B-cell malignancies. At odds with myeloma or WM, the serum concentration of PP is generally low or even undetectable with protein electrophoresis, and its quantity has no impact on clinical features and outcome. Thus, immunofixation or hevylite^TM^ (the binding site, Birmingham UK) tests are necessary to rule out/confirm its presence. Indeed, IgG and IgM are the prevalent PP, while IgA, biclonal, and free-light chains PP are present in a minority of patients (Table 1).

In DLBCL, only the IgM PP has been reported, by almost all authors from the western world, to be a strong prognostic factor, independent from the IPI score, for both PFS and OS [33,34,35,36,38] Noteworthy, Cox et al., reported the detection of recurrent mutations in high impact genes, and the preferential IGVH4-34 gene usage, which allowed them to hypothesize that most IgMs-DLBCL originate from immune-escaped autoreactive B-cells [36]. Indeed, this feature [59] may also explain the high rate of CNS spreading [35,36,38]. Conversely, Johansson et al. [38] found no prevalence of specific gene mutations and of the ABC-type. Therefore, more studies are necessary to shed light on the molecular correlatives and the etiopathogenesis of the IgMs-DLBCL subset.

In low-grade B-NHL and in CLL, the occurrence of gammopathy is strongly associated with adverse features and shorter TFS, while data on PFS and OS are less concordant within different papers. In CLL, the analysis of three large series showed Ig-secreting cases to range between 17% and 35% [21,22,23]. Of note, all authors reported IgM PP to be associated with del17p or TP53 mutations, and IgG PP with trisomy 12, while no association was found with the IGVH mutational status (Table 1). Of note, IgM, IgG, hypo-γ, and normal γ-globulin subgroups exhibited the preferential usage of different IGVH repertoires. Thus, it was speculated that CLL progenitor cells, within these four subgroups, expand through a different type of antigen exposure [22].

In FL, only one large retrospective study investigated the features of PP+ve patients, who represented 25% of the population examined. Gammopathy was increasingly more frequent in older patients, and was significantly associated with POD24, but not with a higher than expected FLIPI score. Of note, in MV analysis, PP+ve was a significant factor for OS, only in patients ≥60 years.

In EMZL, the incidence of PP was reported, in different series, to range from 11% up to 39%, and the prevalent subtype is IgM [46,47,48,50,55,56]. Almost all papers, concordantly reported in PP+ve, compared to PP-ve subjects: older age, more advanced stage, frequent involvement of lymph nodes and of multiple extra-nodal sites, as well as a higher MALT-IPI score [51], Furthermore, Epperla et al., also reported a higher rate of transformation to high-grade and a higher than expected POD24 [55,56]. However, the impact of gammopathy on PFS and OS varies greatly, in different series. This is probably due to the heterogenous systemic treatments in use over the last 20 years. Indeed, in two recent studies, the negative prognostic impact of gammopathy in MV analysis of PFS, was abolished, in the subset of patients who received R-CHEMO [49,56]

In SMZL, the incidence of gammopathy is reported in 20–42% of patients, and it is associated with adverse clinical characteristics and shorter TFS [53,54,56]. However, the advent of R seems to have overcome the adverse outcome of PP+ve patients, compared to the pre-rituximab era [53,56].

Currently, the impact of PP in NHL and CLL patients treated with new targeted therapies, except for rituximab, is almost unknown. Thus, the significance of gammopathy, in the context of novel therapies, needs to be assessed in prospective trials. Also, the underlying molecular profiles and actionable pathways, still await to be thoroughly investigated. Notwithstanding, it is reasonable to speculate that malignant B-cells, which are terminally differentiated and endowed with features of Ig-secreting lympho-plasmocytic cells, may have a more aggressive behavior, and be less sensitive to chemotherapy.

For consideration, not all PP+ve patients have an Ig-secreting B-cell malignancy. In fact, in a minority of cases (range 11–32%), the paraprotein, resulted clonally unrelated to the neoplasia. Indeed, it is currently unknown if also the latter condition negatively impacts patient’s features and prognosis. In fact, some authors speculated that this status, as well as the detection of elevated free-light chains, may reflect immune-dysregulation of the host, which adversely affects patient’s outcome [12,38,43]. Definetely, to confirm that a paraprotein is secreted by the neoplasia, the heavy and light chains, expressed by the malignant B-cell, also need to be characterized. Finally, the value of paraprotein detection and quantification to monitor disease progression and relapse, has been little investigated. Nonetheless, in DLBC, PP seems to be a rather unreliable marker, possibly because the Ig-secretion capability, is lost during clonal evolution [35]. Conversely, in low-grade NHL and CLL, the level of paraprotein was described as a good indicator of disease progression [43]. Indeed, more studies are necessary to fully elucidate its usefulness in disease monitoring.

## 6. Conclusions

Serum paraprotein is associated with adverse features and is an independent prognostic factor for the outcome of patients with CLL, DLBCL, FL, and MZL. Thus, its determination, by immunofixation or hevylite chain assays, which is easy and inexpensive, should become part of the diagnostic workup. However, the biological mechanisms underlying the more aggressive behavior of NHL and CLL, associated with gammopathy, are still almost unknown. Indeed, the relevance of gammopathy in the context of novel targeted therapies, needs to be assessed in prospective studies.

## Figures and Tables

**Figure 1 cancers-15-04440-f001:**
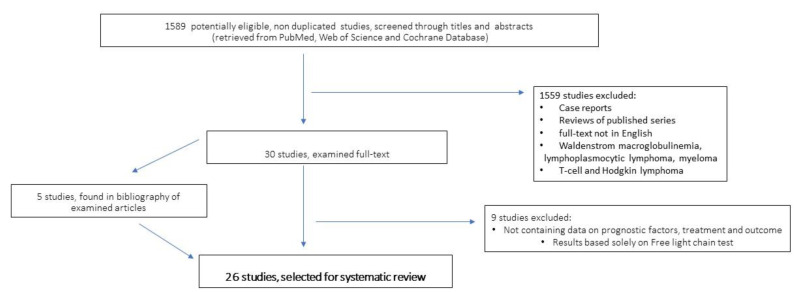
Flow-chart of study selection.

**Table 1 cancers-15-04440-t001:** Clinical features and outcome of mature B-cell malignancies with an associated serum paraprotein.

Disease	PP+ve/All (%)	PP Subtypes (%)	Advanced Stage	*p*	AdversePrognosticFactors	*p*	Outcome	*p*	Reference
CLL	27/133 (20%)	IgM = 44% IgG = 44%	PP+ve vs. ref	<0.001	**17p-/11q--** PP+ve = 67% ref = 33%	0.032	**OS MV** IgG IgM HR = 0.208	ns 0.04	Xu 2011
CLL	222/1505 (17.3%)	IgM = 33% IgG = 67%	IgM = 39.% IgG = 23% hypo-y = 31% ref = 25%	0.002	**17p-** IgM = 15% IgG = 3% hypo-y = 5% ref = 6%	0.022	**TFS MV** IgM HR = 2.187 IgG HR = 1.609 Hypo-y HR = 1.699	<0.0001 0.001 <0.0001	Corbingi 2020
							**OS MV**	ns	
CLL	52/150 (34.6%)	IgM = 52% IgG = 48%	IgM = 63% IgG = 37.5% hypo-y = 22% ref = 3.5%	<0.0001	**17p-/11q-/+12** IgM = 60% IgG = 50% Hypo-y = 19% ref = 23%	<0.001	**TFS MV** IgM HR = 2.63 IgG HR = 3.55 Hypo-y HR = 2.34	0.0031 0.0002 0.0059	Rizzo 2015
							**OS MV**	ns	
DLBCL	158/492 IgM = 102 IgG = 52 IgA = 3 Other = 1		IgM = 83% O-PP = 68% ref = 65%	0.002	**IPI = 3–5** IgM = 55% O-PP = 40% ref = 36% **CNS-IPI = 4–6** IgM = 36% O-PP = 14% ref = 15%	<0.001 <0.001	**PFS MV** IgM, HR = 3.62 O-PP, HR = 1.54	<0.001 ns	Cox 2022
							**OS MV** IgM, HR = 3.57 O-PP, HR = 1.53	<0.001 ns	
DLBCL	154/409 (37.6%)	IgM = 51.6% IgG = 17% IgA = 8% LC = 24.5%	IgM vs. ref LC vs. ref O-PP n.a.	0.002 0.001	**IPI-score 3–5** IgM vs. ref LC vs. ref O-PP vs. ref **Age-adjusted IPI = 2–3** IgM vs. ref LC vs. ref O-PP n.a.	<0.001 0.008 n.a. 0.002 0.013	**PFS MV** IgM HR = 1.53 O-PP	0.032 ns	Jardin 2013
							**OS MV** IgM O-PP	ns ns	
DLBCL	19/108 (18%)	n.a.	ref = 51% IgM = 74%	0.079	**IPI-score = 3–5** IgM vs. ref	0.025	**PFS MV** IgM, HR = 5.44 **OS MV** IgM, HR = 8.15	<0.001 <0.001	Johansson 2023
FL	82/299 (27%)	IgM = 25.6% IgG = 50% IgA = 5% LC = 11% Biclonal = 9%	PP vs. ref	ns	Extranodal sites Beta2 > N POD24	0.05 0.002 0.02	**PFS (10-years)** ref vs. PP+ve PFS MV	0.0076 ns	Mozas 2021
							**OS (10-years)** ref vs. PP **OS MV** All PP+ve vs. ref >60y PP+ve vs. ref	0.045 ns 0.02	
EMZL (non gastric)	36/208 (17%)	IgM: 66% IgG: 25% IgA: 8%	n.a.		n.a.		OS MV (HR n.a.)	0.04	Arcaini 2006
EZML	42/176 (19.3%)	IgM = 64% IgG = 14.3% Biclonal = 12% LC = 2.4%	PP 74% vs. ref 29%	<0.0001	Extranodal > 2 Beta2 > N Nodal involvement MALT-IPI	<0.0001 <0.0001 <0.0001 0.001	**PFS MV** PP+ve, HR = 2.31 **OS MV** PP+ve, HR = 4.14	0.018 0.026	Ren 2022
EZML	35/328 (10.7%)	IgM = 43% IgG = 37% IgA = 14% Biclonal = 3% LC = 3%	n.a.		n.a.		**PFS (5-years)** ref vs. PP+ve **PP+ve PFS MV**	0.015 ns	Alderuccio 2019
							OS (5-years) ref vs. PP **OS MV**	0.038 ns	
MZL EMZL NMZL SMZL	173/547 (32%) 86/319 (50%) 49/121 (28%) 38/107 (22%)	IgM = 56% IgG = 31% IgA = 5% Biclonal = 3% LC = 5%	PP 73% vs. ref 54%	0.01	Age (median) BM infiltration PS Ecog HT POD24	<0.01 0.001 0.01 0.001 <0.0001	**TFS** ref vs. PP+ve	ns	Epperla 2023 *Blood adv* & *J Hematology & Oncol*
							**PFS MV** All PP+ve HR = 1.74 R-Benda PP+ve	0.004 ns	
							**OS MV PP+ve**	ns	
SMZL	34/81 (42%)	IgM 43% IgG 37% IgA 3% biclonal 2%	n.a.		n.a.		**TTP MV PP+ve**	<0.006	Thieblemont 2002
							**OS MV PP+ve**	<0.001	

This study was enriched with PP+ve cases. Thus, the rate of IgM and O-PP, is not representative of PP incidence. LC = light chain alterations; PP = paraprotein; PP+ve = presence of serum paraprotein; PP-ve = absence of serum paraprotein; O-PP = other paraprotein subtypes; HT = histologic transformation to high-grade NHL. DLBCL = diffuse large B-cell lymphoma, CLL = chronic lymphatic leukemia; SMZL = splenic marginal zone B-cell lymphoma; EMZL = extranodal marginal zone lymphoma; NMZL = nodal marginal zone lymphoma; FL = follicular lymphoma; OS = overall survival; TFS = treatment-free survival; PFS = progression-free survival; MV = multivariate analysis.

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
