# Peer review of "Serum Paraprotein Is Associated with Adverse Prognostic Factors and Outcome, across Different Subtypes of Mature B-Cell Malignancies—A Systematic Review"

_cancers, 2023, doi:10.3390/cancers15184440_

Round 1

Reviewer 1 Report

The authors summarize evidence for circulating paraproteins in non-hodgkin lymphoma and the potential application as a biomarker for prognosis and risk-stratification. The review is overall balanced, thorough, and well written. There are some grammatical errors and redundancy in use of English terminology that needs to be edited to improve the quality of the paper.

The authors summarize evidence for circulating paraproteins in non-hodgkin lymphoma and the potential application as a biomarker for prognosis and risk-stratification. The review is overall balanced, thorough, and well written. There are some grammatical errors and redundancy in use of English terminology that needs to be edited to improve the quality of the paper.

Author Response

  • The authors summarize evidence for circulating paraproteins in non-hodgkin lymphoma and the potential application as a biomarker for prognosis and risk-stratification. The review is overall balanced, thorough, and well written. There are some grammatical errors and redundancy in use of English terminology that needs to be edited to improve the quality of the paper.

The English language has been fully revised by a mother-tongue English Teacher

Reviewer 2 Report

A systematic review of B malignancies-associated PPs is worth the effort for publication but some methodological issues related to the criteria used for studies selection and interpretation should be addressed before considering your work for publication.

1.       In “Methods and criteria of bibliographic research and articles selection” temporal criteria of research and article language should be specified;

2.       Different databases from PubMed should be interrogated;

3.       A more extended range of keywords should be used in order to capture a greater number of papers (e.g. “free light chain”, “lymphoma”, etc.). I tried by myself a quick research using the keywords “chronic lymphocytic leukemia”, “lymphoma” and “free light chain” and found out at least two works regarding the prognostic impact of monoclonal FLC on CLL/lymphoma that were not included by the Authors (one of these is: Am J Hematol. 2014 December ; 89(12): 1116–1120. doi:10.1002/ajh.23839);

4.       What kind of inclusion and exclusion criteria have been used? Were the papers reviewed by all authors? A flow chart of paper selection might be provided;

5.       Analytical issues regarding the interpretation of protein serum assays should be addressed and discussed throughout the paper. For example, the use of FLC ratio is considered a poor marker of clonality (Singh G. Serum Free Light Chain Assay and κ/λ Ratio Performance in Patients Without Monoclonal Gammopathies:  High False-Positive Rate. Am J Clin Pathol. 2016 Aug;146(2):207-14. doi: 10.1093/ajcp/aqw099. PMID: 27473738). You should explicitly indicate whether papers using nonspecific markers have been included or not in your analysis and why;

I think that if you considered these issues and decided to address them, the bibliographic research would  likely be more representative of the topic. It is probable that the whole paper should be revised by including  undetected studies and excluding papers of low quality. Another round of revision would then be reasonable.

Author Response

2) A systematic review of B malignancies-associated PPs is worth the effort for publication but some methodological issues related to the criteria used for studies selection and interpretation should be addressed before considering your work for publication.

  1. In “Methods and criteria of bibliographic research and articles selection” temporal criteria of research and article language should be specified;

This has been addressed:  by providing a flow-chart (figure 1) ans a section where the search results are reported

  1. Different databases from PubMed should be interrogated;

This was addressed in the methods section

  1. A more extended range of keywords should be used in order to capture a greater number of papers (e.g. “free light chain”, “lymphoma”, etc.). I tried by myself a quick research using the keywords “chronic lymphocytic leukemia”, “lymphoma” and “free light chain” and found out at least two works regarding the prognostic impact of monoclonal FLC on CLL/lymphoma that were not included by the Authors (one of these is: Am J Hematol. 2014 December ; 89(12): 1116–1120. doi:10.1002/ajh.23839);

This has been addressed. Actually,  papers in which solely FLC test was used to assess gammopathy, were excluded from our  review. Notwithstanding a full explanation (and references) of this choice is given  the introduction  “The routine screening tests, used in clinical practice to detect paraproteinemia (PP), in serum and urine, are:  protein electrophoresis (PE) and immune-electrophoresis (IFE) or heaylite chain assay  (HLC, hevyliteTM, the Binding site, Birmingham UK), plus serum free-light-chain test (sFLC, FreeliteTM, the Binding site, Birmingham UK). The latter, allows to quantify k and λ light chains concentration and the k/λ ratio,  whichare useful for both disease prognostication and monitoring. In addition, sFLC is pivotal to diagnose light-chain diseases, However, the latter cannot stand alone in the diagnostic work-up of PP. In fact, in ≥30% of patients with paraproteinemia, there is a normal k/λ ratio  (Singh, 2016) (Singh, 2016) , while an abnormal k/λ chains ratio is present in up to 36% of subjects, without monoclonal lymphoproliferation (Singh, 2016, 2020)   . Indeed to NHLFor all the above, we did not include in our analysis, studies which relied only on the sFLC test ((Maurer 2013; Maurer, Cerhan, et al., 2011; Maurer, Micallef, et al., 2011; Morabito et al., 2011; Pratt et al., 2009; Witzig et al., 2014) “

  1. 4.What kind of inclusion and exclusion criteria have been used? Were the papers reviewed by all authors? A flow chart of paper selection might be provided;

This is described now in the methods section

  1. 5.Analytical issues regarding the interpretation of protein serum assays should be addressed and discussed throughout the paper. For example, the use of FLC ratio is considered a poor marker of clonality (Singh G. Serum Free Light Chain Assay and κ/λ Ratio Performance in Patients Without Monoclonal Gammopathies:  High False-Positive Rate. Am J Clin Pathol. 2016 Aug;146(2):207-14. doi: 10.1093/ajcp/aqw099. PMID: 27473738). You should explicitly indicate whether papers using nonspecific markers have been included or not in your analysis and why;

As detailed above this issue has been addressed and relevant references included

I think that if you considered these issues and decided to address them, the bibliographic research would  likely be more representative of the topic. It is probable that the whole paper should be revised by including  undetected studies and excluding papers of low quality. Another round of revision would then be reasonable.

Reviewer 3 Report

It is a very well written and comprehensive manuscript focusing on prognostic significance of paraproteinemia in patients with low nad high grade B-cell NHL. 

Overall the data are very well presented, however below there are minor comments for further improving the quality of the nmanuscript.

1) Manuscript should be condensend, especially the discussion where many data are repeated. I propose discussion should include only the general conclusions and or comments

2) Since the presence of MGUS is common especially in the elderly, authors should clarify if in these studies paraprotein were produced by the same clone as the lymphoma. They can test if the light chain for example of serum paraprotein was also expressed on the lymphoma cell

3) Authors should present more data if possible regarding the elimination of Paraprotein after successful treatment and if there is evidence of reccurence after relapse of the lymphoma

4) Since it seems that the presence of paraprotein is a negative prognostic factor in almost all presented studies, it will be important if authors speculate about the relationship between paraprotein and the outcome  

Author Response

3)It is a very well written and comprehensive manuscript focusing on prognostic significance of paraproteinemia in patients with low nad high grade B-cell NHL. 

Overall the data are very well presented, however below there are minor comments for further improving the quality of the nmanuscript.

  • Manuscript should be condensend, especially the discussion where many data are repeated. I propose discussion should include only the general conclusions and or comments

In agreement with this good suggestion, the discussion section has been substantially synthetized

  • Since the presence of MGUS is common especially in the elderly, authors should clarify if in these studies paraprotein were produced by the same clone as the lymphoma. They can test if the light chain for example of serum paraprotein was also expressed on the lymphoma cell

This issue is now addressed in the results section (DLBCL and FL) and in the final part of the discussion section

  • Authors should present more data if possible regarding the elimination of Paraprotein after successful treatment and if there is evidence of reccurence after relapse of the lymphoma

Unfortunately, there is little data on this issue. However both in the result section (DLBCL and FL) and in the final part of the discussion this point is now debated

  • Since it seems that the presence of paraprotein is a negative prognostic factor in almost all presented studies, it will be important if authors speculate about the relationship between paraprotein and the outcome . 

This is a very good suggestion. We have made an hypothesis, in the last part of the discussion section.